# A 31-Inch AMOLED Display Integrating a Gate Driver with Metal Oxide TFTs

**DOI:** 10.3390/mi16121325

**Published:** 2025-11-26

**Authors:** Xianjie Zhou, Qiming Zeng, Li Guo, Yicheng Yu, Fei Yu, Guo Tian, Xiaopeng Lu, Zhiqiang Zhang, Baixiang Han, Yan Xue

**Affiliations:** 1School of Electronic and Communication Engineering, Shenzhen Polytechnic University, Shenzhen 518055, China; 2Shenzhen Zhongchengda Applied Materials Co., Ltd., Shenzhen 518000, China; 3Shenzhen China Star Optoelectronics Semiconductor Display Technology Co., Ltd., Shenzhen 518000, China

**Keywords:** AMOLED, GOA, IGZO-TFTs, process stability

## Abstract

Gate driver-on-array (GOA) circuits employing amorphous indium–gallium–zinc oxide (IGZO)-based thin-film transistors (TFTs) have been successfully utilized to generate the driving signals for the commercialization of active-matrix organic light-emitting diode (AMOLED) displays. The depletion-mode TFTs in GOA circuits can be completely turned off by the introduction of series-connected, two-transistor, dual low-voltage-level power signals. Simulation results demonstrate that a GOA exhibits high process stability with a threshold voltage margin from −5 V to +5 V. Furthermore, the GOA output characterization and mobility compensation effect are evaluated by the integration of the GOA and pixel in a 31-inch 4K AMOLED display. Experimental results demonstrate that full-swing driving pulses can be obtained with the GOA. Finally, the stripe mura in the display caused by mobility variation can be successfully eliminated by the introduction of GOA circuits.

## 1. Introduction

Along with the rapid development of displays with improved lightness and wearability, the active-matrix organic light-emitting diode (AMOLED) has become an emerging technology potentially exhibiting the advantages of small volume and weight, a fast response, and a wide color gamut [1,2,3,4]. Meanwhile, since this technology is highly compatible with the ink-jet printing process, a growing number of flat-panel display manufacturers have proposed an AMOLED roadmap [5]. Nevertheless, the manufacturing cost has been the major challenge to the commercialization of AMOLED displays; therefore, the utilization of AMOLED displays is limited to the premium consumer electronic market worldwide [6]. A high demand for low-cost thin-film transistor (TFT)-compatible, hybrid integrated circuit systems with a simplified manufacturing process has arisen. In these systems, the direct fabrication method of gate driver-on-array (GOA) circuits enabled by the TFT process is utilized to generate gate pulse signals instead of external gate driving chips, and the related gate chip bonding process can be simplified [7,8].

Two well-established TFT techniques have been explored to fabricate AMOLED backplanes, i.e., low-temperature poly-Si (LTPS) [9,10] and amorphous indium–gallium–zinc oxide (IGZO) [11,12]. LTPS TFTs present excellent electrical properties that are characterized by their high mobility (~100 cm^2^/V·s) and good high-temperature reliability (stable at a temperature of 60 °C), while a complex process using an excimer laser annealing (ELA) treatment is required to improve the crystalline effect in poly-Si films [13]. The poor crystalline uniformity of Si films during the ELA process and the high ELA equipment cost limit the use of LTPS TFTs in high-generation display manufacture lines [14,15]. The recently developed IGZO films that serve as channel materials in active layers demonstrate high uniformity on a large scale [16,17]. Especially, IGZO-TFTs have been utilized in the commercial application of AMOLED displays in the mainstream generation 8.5 line (glass size = 2200 mm × 500 mm) [18].

The successful use of IGZO-TFTs in AMOLED displays has motivated the research on the GOA with IGZO-TFTs, particularly the circuit topology design with depletion-mode TFTs [19,20]. Noteworthily, a recent work performed by LG’s researchers implies that IGZO-TFTs exhibit poor temperature reliability [21,22]. In their research, when displays are turned on, the transformation of operation temperature in displays causes brightness mura, rendering image quality sensitive to variations in mobility in TFTs. To enable the adoption of IGZO-TFTs in AMOLED displays, a pixel design with a mobility compensation function must be introduced to eliminate the brightness mura across the displays, and it is necessary for the IGZO-GOA to export high-speed shift register signals, as well as driving pulses for the mobility compensation function. However, AMOLED displays with the GOA technology have not yet entered the mass production phase because most studies on the IGZO-GOA focus on the conventional shift register programing mode [23,24,25].

In this work, we demonstrated an external mobility compensation system for AMOLED displays, in which an IGZO-TFT-based GOA circuit was utilized to export shift register signals and external mobility compensation signals simultaneously. Subsequently, the mechanisms of external compensation technology and GOA topology were systematically investigated. Furthermore, the performance and reliability of GOA were evaluated by integrating the GOA into a 31-inch 4K AMOLED display. Finally, the mobility compensation procedure was performed to improve the image quality.

## 2. The Mechanism of the External Compensation System

The ability to detect the mobility information of driving TFTs in pixels is fundamental for the commercialization of AMOLED displays based on the IGZO-TFT technology. In order to maintain high picture quality, an external compensation system, shown in Figure 1, was utilized to avoid the influence of mobility on luminance over the whole display. In this structure, a basic pixel cyclic unit is composed of three sub-pixels (R/G/B). According to the display resolution (2160 × 3840, 4K), an individual pixel is repeatedly expanded toward the horizontal and vertical directions. The accurate placement of sixteen source chips on the bottom side supplies significant display signals for this system. Each source chip can support 720 data signals and 720 sense signals, corresponding to the number of data and sense lines in 2160 × 240 pixels. The overall data signals are timing-controlled and image-processed by two separate time-controller boards, which are synchronized with each other so that no image discrepancy occurs between the left and right sides of the display. Meanwhile, GOA circuits are arranged symmetrically on both sides to provide the driving signals for pixels.

The time diagram of convenient GOA is provided in Figure 2a. The displays do not perform any special operations in the blank time. The pixels in each single row emit light immediately after the data voltage is transferred to the pixels in the system. Shift register pulses, separated by the blank time at the end of each frame, transport to the display area row by row [26,27,28]. As shown in Figure 2b, the progressive emission programming scheme was used in our proposed system. The duration of blank time (T*_blank_*) is expressed as follows:(1)$$T_{blank}=\frac{1}{F}-h\times t$$

*t* (1.85 μs) denotes the charging time of each pixel, *h* (2160) denotes the number of rows on the display, and *F* (120 HZ) denotes the frequency. The blank time (170 μs) is calculated through the above formula and can be utilized to transmit closed captions, program rating information, time codes, and other data. The function of mobility compensation is to reduce the carrier mobility differences in TFTs in AMOLED pixels, thereby preventing the inconsistent light-emitting brightness among different pixels. The mobility compensation is performed during the blank time period to avoid affecting the effective display and ensure real-time compensation because these operations demand an independent timing window that does not interfere with the display.

The schematic of external compensation system is provided in Figure 3a, and each pixel circuit is composed of two switching TFTs (T2, T3), a driving TFT (T1), and a storage capacitor (Cst). Two scan signals (WR, RD), generated by the GOA, are used to supply switching signals. Simultaneously, data voltage is utilized to control the gray scale in pixels. Furthermore, a high constant voltage (VDD, 24 V) and a low constant voltage (VSS, 0 V) are used to control the drain electrode voltage of T1 and the cathode voltage of OLED. The establishment of 3T1C circuit ensures that the mobility message of T1 can be detected by an external sense system. In this system, the anode of OLED, referred to as S in the circuit, connects to a source chip through the metallic sense lines on glass. Each of the Spre and Sam switches alternatively performed a turn-on operation and a turn-off operation in the source chips. The reference source (*Vref*, 1 V) and analog-to-digital converter (ADC) are applied to reset the voltage in S and transfer the voltage message to a digital signal, respectively.

Figure 3b shows the time diagram of individual pixel at the emission stage. The operation can be divided into two stages. At first, WR and RD are simultaneously switched to a high voltage. Then, T2 and T3 are turned on, and the data voltage (*V_data_*_0_, 1~5 V) and *Vref* are separately written to the gate electrode and source electrode of T1. T1 works in the saturation regime because the *Vds* (*VDD*-*Vref*) voltage is profoundly higher than the Vgs − *Vth* (*V_data_*_0_ − *Vref* − *Vth*) voltage, in which *Vth* represents the threshold voltage. The drain current of T1 is expressed as follows:(2)$$I_{\mathrm{ds}0}=\frac{1}{2}\mu C_{ox}\sqrt{\frac{W}{L}}{\left(V_{\mathrm{data}0}-Vref-Vth\right)}^{2}$$
where *W* represents the channel width, *L* represents the channel length, Cox represents the insulation capacitance between active layer and gate electrode, and *μ* represents the initial mobility. At this time, the lower voltage of *Vref* (1 V) than the threshold voltage of OLED (2.5 V) ensures that the OLED is completely turned off. Secondly, when WR and RD are discharged at a low voltage, T2 and T3 are switched off. Simultaneously, the voltages of G and S are coupled interactively to a high voltage until the OLED is turned on. Theoretically, the current penetrating through the OLED is maintained in accordance with *I* in Equation (2). As the display starts to work, the operation temperature increases significantly and the drain current begins to deviate from the raw data in the system. It is necessary to perform the mobility compensation operation during the display time.

The blank time period is immediately adjacent to the display of the next frame for real-time compensation. Figure 3c indicates the procedure of mobility compensation at the blank time, which is divided into five stages: In the first stage, RD and Spre are switched to a high voltage. The OLED device is turned off, since the voltage of S is reset to a low voltage *Vref*. In the next stage, T2 is turned on by switching WR from a low voltage to a high voltage. The voltage of G is written as fixed data (*V_data_sense_*). The current flowing through T1 is as follows:(3)$$I_{\mathrm{ds}1}=\frac{1}{2}\left(\mu +\Delta \mu \right)C_{ox}\sqrt{\frac{W}{L}}{\left(V_{data\_sense}-Vref-Vth\right)}^{2}$$

Δ*μ* represents the variety of mobility. In the third stage, WR and Spre are switched to a low voltage, and then the current flows toward the external source chips through T3. Equation (4) shows the electron transfer process at this time:(4)$$I_{\mathrm{ds}1}\times t=C_{sense}\times \left(V_{s}-Vref\right)$$

*t* denotes the charging time, and *C_sense_* denotes the parasitic capacitance of sense line. The voltage in S (*Vs*) recorded at the end of this stage can be detected using the external sensing unit. Obviously, the relationship between *Vs* and mobility (*μ +* Δ*μ*) can be established. Before the display is turned on, the initial drain current *I_ds_*_0_ can be expressed as follows:(5)$$I_{\mathrm{ds}0}=\frac{1}{2}\mu C_{ox}\sqrt{\frac{W}{L}}{\left(V_{data\_sense}-Vref-Vth\right)}^{2}$$(6)$$I_{\mathrm{ds}0}\times t=C_{sense}\times \left(V_{s0}-Vref\right)$$

Assuming mobility is well compensated (*I_ds_*_0_ = *I_ds_*_1_), it is easy to calculate the mobility message based on Equations (3)–(6).(7)$$\frac{\mu +\Delta \mu }{\mu }=\frac{V_{s}-Vref}{V_{s0}-Vref}$$

It is evident that the mobility (*μ +* Δ*μ*) can be characterized through the variation in voltage in S. In the fourth step, Sam is switched on, and then Vs transfers from the analog signal to the digital signal through the ADC. In the final step, T2 and Spre are turned on, and a new data signal is written to pixel (*V_data_*_1_) in the next frame. In order to obtain the same current as in Equation (2), *V_data_*_1_ is calculated using the following equation:(8)$$\frac{\left(\mu +\Delta \mu \right)}{\mu }=\frac{{\left(V_{\mathrm{data}0}-Vref-Vth\right)}^{2}}{{\left(V_{\mathrm{data}1}-Vref-Vth\right)}^{2}}$$

According Equations (7) and (8), *V_data_*_1_ is expressed as Equation (9):(9)$$V_{\mathrm{data}1}=\sqrt{\frac{V_{s0}-Vref}{V_{s}-Vref}}(V_{\mathrm{data}0}-Vref-Vth)+Vref+Vth$$

*V_data_*_1_ is written to the pixel in the next fame. The drain current of T1 in the next frame is expressed as follows:(10)$$I_{ds\_new}=\frac{1}{2}\left(\mu +\Delta \mu \right)C_{ox}\sqrt{\frac{W}{L}}{\left(V_{\mathrm{data}1}-Vref-Vth\right)}^{2}$$

According to Equations (9) and (10), the current is equal to the current in the previous frame after compensation, as shown in Equation (11):(11)$$I_{ds\_new}=I_{\mathrm{ds}0}$$

Therefore, the brightness non-uniformity caused by mobility differences can be promptly corrected by calculating the compensation value based on the data (*V_data_*_0_) from the previous frame and writing the simulated data (*V_data_*_1_) to the pixel in the next frame, preventing the accumulation of image deviations.

## 3. The Operation of GOA Circuit

Recent research performed by B. Kim et al. indicates that IGZO-TFTs often act as depletion-mode devices [29]. The direct utilization of traditional GOA circuit may cause the malfunction of circuit. A series-connected, two-transistor (STT) structure has shown the potential to reduce the leakage in TFTs by introducing a feedback unit in the circuit [30]. Meanwhile, a dual low-voltage-level power (VGL) structure is used to reduce the leakage current from TFTs, which primarily serves to decrease the gate-to-source voltage (Vgs). Figure 4 exhibits the schematic and time diagram of the proposed GOA circuit, which is composed of a pull-up control (T11, T12) unit, pull-up (T21~T23) unit, pull-down unit (T31~T34, T71~T74), pull-down holding (T41~T410), inverter unit (T51~T58), feedback unit (T6), logical sense unit (T81~T85), and bootstrap capacitor (Cbt1~Cbt4). In this circuit, the STT structure is used to decrease the leakage current of TFTs connected to Q. Meanwhile, in order to improve the driving force in scan signals (WR, RD), a dual VGL structure is utilized to decrease the leakage current in T41~T44. LC1 and LC2 are a group of low-frequency AC signals with an opposite voltage phase, while CKa, CKb, and CKc are a group of high-frequency AC signals.

The operation of the proposed unit can be divided into the following periods:

(1) Program time: In the P1 stage, the carrying signal Cout(n − 6) switches from a low voltage to a high voltage. At this time, Q is charged to a high voltage through T11 and T12. Cout(n), WR(n), and RD(n) are set to a low voltage when T21, T22, and T23 are turned on. LC1 is at a high voltage, and the first inverter unit (T51, T53, T55, T57) is utilized to reverse the voltage between QB1 and Q. Simultaneously, QB2 maintains a low voltage and T41~T410 are turned off. LSP is switched to a high voltage, and then T81 and T82 are turned on. Subsequently, M rises to a high voltage. In the P2 stage, Cout(n − 6) and LSP are switched to a low voltage, T11, T12, T81, and T82 are turned off, and Q and M maintain a high voltage. In the P3 stage, CKa, CKb, and CKc are switched to a high voltage. Consequently, in contrast to the voltage in phase P1, Q is bootstrapped to a higher voltage through Cbt1~Cbt3. Cout(n), WR(n), and RD(n) are switched to a high voltage. In the P4 stage, CKa, CKb, and CKc are switched to a low voltage. Q is bootstrapped to a high voltage through Cbt1~Cbt3. Cout(n), WR(n), and RD(n) are released at a low voltage. In the P5 stage, Cout(n + 6) is changed from a low voltage to a high voltage, T31 and T32 are turned on, Q is released at a low voltage, and QB1 rises to a high voltage. At this moment, T42, T44, T46, T47, and T48 are turned on, and Q, Cout(n), WR(n), and RD(n) maintain a low voltage. In the P6 stage, Cout(n + 6) switches to a low voltage, and then T31 and T32 are turned off.

(2) Blank time: In the B1 stage, M maintains a high voltage, and Reset1 is changed from a low voltage to a high voltage. Since T84 is turned on, Q is charged to a high voltage (VGH) and QB1 transfers from a high voltage to a low voltage. Meanwhile, Cout(n), WR(n), and RD(n) are set to a low voltage. In the B2 stage, CKc transfers from a low voltage (VGL1) to a high voltage (VGH), due to the existence of Cbt1. Q is theoretically bootstrapped to a high voltage VGH + (VGH-VGL1)Cbt1/(Cbt1 + Cbt2 + Cbt3), and then RD(n) is charged to a high voltage. In the B3 stage, CKb switches from a low voltage to a high voltage, and Q is bootstrapped to VGH + (VGH-VGL1)(Cbt1 + Cbt2)/(Cbt1 + Cbt2 + Cbt3). WR(n) is charged to a high voltage. In the B4 stage, since CKb changes from VGH to VGL, WR(n) is discharged from VGH to VGL. At this moment, Q is decreased to VGH + (VGH-VGL1)Cbt1/(Cbt1 + Cbt2 + Cbt3). In the B5 stage, WR(n) is charged to a high voltage due to the transformation from a low voltage to a high voltage in CKb. In the B6 stage, CKb and CKc are at a low voltage. WR(n) and RD(n) are discharged at VGL. In the B7 stage, LSP and Reset2 change to VGH. Q is discharged at a low voltage through T33 and T34. Simultaneously, QB1 rises to a high voltage, and T42, T44, T46, T47, and T48 are turned on. Simultaneously, M is changed from VGH to VGL through T81 and T82. In the B8 stage, LSP and Reset2 switch to a low voltage, and T81, T82, T33, and T34 are turned off.

To improve the reliability in the GOA, an investigation to introduce the two-group inverter unit and pull-down unit is performed in this study. This is necessary for decreasing the positive bias temperature stress (PBTS) effect in the TFTs and improve the electric reliability of TFTs [31]. The schematic of the conventional pull-down unit and the time diagram of the conventional pull-down unit are provided in Figure 5a and Figure 5b, respectively. It is evident that the voltage in QB is mostly maintained at a high voltage. Therefore, T41~T44 continuously undergo PBTS during the working time. In this study, the dual pull-down unit has been utilized in GOA circuits. The schematic of the proposed pull-down unit and the time diagram of the conventional pull-down unit are provided in Figure 5c and Figure 5d, respectively. The voltage of LC1 and LC2 switches between high and low for an interval of 100 frames. Alternately, T41~T44 and T45~T48 undergo PBTS during the working time. Therefore, the introduction of the two-group inverter unit and pull-down unit can reduce the PBTS effect and improve reliability.

Figure 6a,b exhibit the GOA diagrams and the waveforms of CKs, respectively. Noteworthily, CK signals are relevant for the GOA circuits in two ways. At first, the increasing number of CK lines reduces the signal delay time in terms of the significant RC loading in displays with large sizes. More importantly, an overlapped structure in these CKs has been successfully developed to generate output waveforms in the GOA circuits with high quality. Considering the balance between RC loading and rise time, the pulse widths of CK signals are settled to 8.9 μs during the program time, and the GOA circuits are driven by 36 CKs (a1~a12, b1~b12, c1~c12), which are supplied by the bypass pins from external source chips. In this display, the whole GOA circuits are composed of 181 basic loop units (12 GOA stages in a basic unit). Furthermore, an additional unit (the 181th unit), the output of which must be disconnected from scan lines in the display area, is used to supply feedback signals for the 180th GOA unit. In addition, Cout(n − 6) signals in the first six stages are driven by the external start pulse.

## 4. GOA Circuit Simulation and Operation

The smart spice simulation was performed to study the function of the GOA circuit. Table 1 and Table 2 list the physical sizes of devices in the GOA circuit and the amplitudes of input signals, respectively. It is evident that the TFT size varies from 10 μm/8 μm to 2500 μm/8 μm. Due to the significant difference in the I-V performance between TFTs with various sizes, it is necessary to extract an individual model card for each TFT size combined with its I-V characterizations. In order to improve the accuracy of the simulation results from spice, the fitting curves of models must highly coincide with those from their initial experiment results.

The spice simulation results in Figure 7a show the output waveform of the 2160th GOA stage, which is smooth and clear without any distortion. Moreover, the GOA exhibits good stability, which is confirmed by the output waveform when the threshold voltage in TFTs shifts to −5 V. Nevertheless, when Vth shifts by −6 V, the high voltage is released in Q and a malfunction takes place in the GOA. When Vth shifts by 6 V, the charging and discharging speeds decrease, resulting in the distortion of the output waveform, shown in Figure 7b.

Figure 8 shows the layout of GOA with a width of 9.2 mm, which is fabricated through the top-emission AMOLED manufacturing process in a G4.5 experimental line. To investigate the GOA function, an oscilloscope was utilized to assess the output waveforms from the GOA, and its results are shown in Figure 9. During this frame, the system is set to carry out the mobility compensation operation in the first stage of GOA, which can be successfully detected at the blank time in Figure 9a. Meanwhile, the measured output waveform of the 2160th stage is shown in Figure 9b. The GOA yields rising time (RT) and falling time (FT) of 5.2 μs and 3.6 μs for the left side of the display and 5.5 μs and 3.8 μs for the right side, respectively. Theoretically, the charging rate is equal to the discharging rate for a stable signal source.

As a measure to maintain comparability, RT must be modified to the same level with FT (from −6 V to 20 V) because the potential difference at the rise stage is higher. After voltage modification, RT is decreased to 4.3 μs for the left GOA and 4.5 μs for the right GOA. The slight difference between RT and FT is due to the potential driving force difference during the charging (0.15 μs) stage and the discharging (0.1 μs) stage in external chips. The resistive load and capacitance load extracted from the simulation tool for each scan line are 4.1K and 850 pF (τ = 3.5 μs), respectively. Besides the scan line, the RC loading in the CK line is another important factor that affects the transport speed of signals. Depending on simulation result, the resistive load and capacitance load for each CK line are 0.26K and 884 pF (τ = 0.22 μs), respectively. It can be easily concluded that the total delay time (τ = 3.72 μs) from the simulation results fits well with the experimental FT result, illustrating that the discharging rate is fast enough to release the voltage in scan lines.

Figure 10 shows the output amplitude as a function of stress time when the GOA works under the condition of high temperature and humidity (60 °C and 90% R.H.). The display continues to work with a checkerboard pattern. According to the display industry standard, a minimum operation time of 500 h is required for mass production. The GOA waveform indicates full-swing output voltage with the operation time increasing to 480 h. Meanwhile, the display demonstrates a high-quality chessboard image without any dark spots or shrinkages in pixels. Although the output amplitude deteriorates when the operation time is increased to 672 h, a highly uniform picture can be still obtained. Finally, when the operation time reaches 696 h, the malfunction of GOA occurs, which is consistent with the picture of the display provided in Figure 10. Therefore, a maximum lifetime of 672 h is obtained, demonstrating the high stability of GOA circuits.

It has been found out that TFTs can gradually recover to their initial performance after an electric stress operation [32,33]. Since the display is turned off and stored at room temperature for 2 h, the output voltage recovers from 5.7 V to 8.5 V, illustrating that the malfunction of GOA is caused by the deterioration of TFT performance. When the display is turned on, the image in the display starts to deteriorate. The working temperature, total current, luminance, and OLED efficiency generated by the display increase monotonically with the operation time. The observation of stripe mura in Figure 11a suggests that the picture is affected by the variety of mobility in TFTs. External compensation is performed to eliminate the influence of mobility difference on image quality. Obviously, a high-quality display image is obtained after mobility compensation, as shown in Figure 11b.

## 5. Conclusions

We demonstrated the feasibility of utilizing the mobility compensation technology employing GOA circuits in AMOLED displays. The working mechanisms of mobility compensation method and GOA technology were systematically investigated. Through the integration of a logical sense unit into the GOA, driving pulses for the mobility compensation process in pixels were generated in scan signals at the blank time. Moreover, due to the utilization of STT and dual VGL power signals, the depletion-mode TFTs were completely turned off and the GOA exhibited high stability with the Vth process margin from −5 V to 5 V. The experimental results confirm that sequential pulse signals can be periodically transferred stage by stage in the GOA. Finally, the image quality of the display is improved significantly by the introduction of mobility compensation technology.

## Figures and Tables

**Figure 1 micromachines-16-01325-f001:**
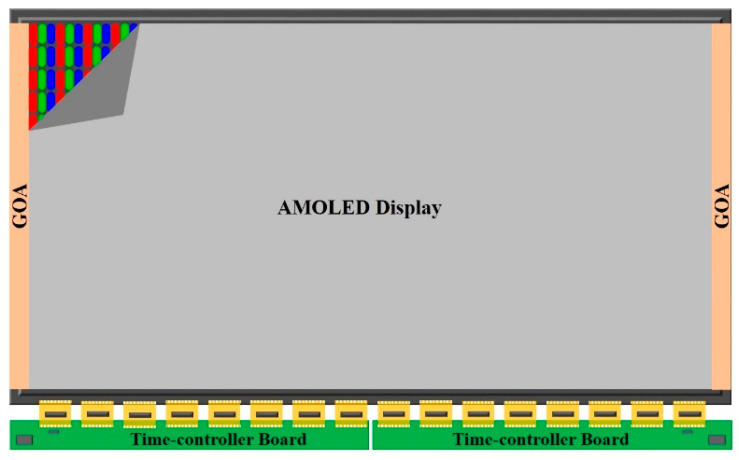
The outline of external compensation system in displays.

**Figure 2 micromachines-16-01325-f002:**
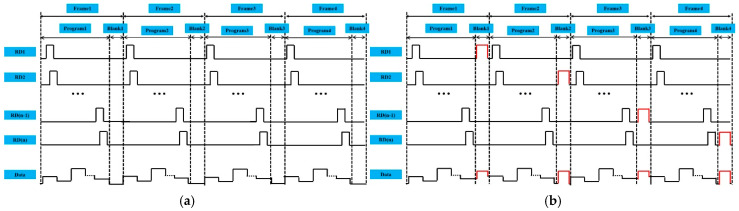
The time diagrams of (**a**) normal scan signal and (**b**) scan signal with the mobility compensation function.

**Figure 3 micromachines-16-01325-f003:**
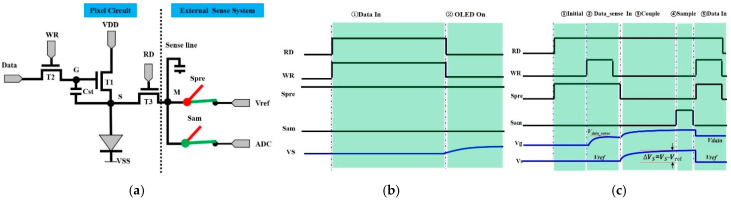
(**a**) A schematic of external compensation system; (**b**) the timing diagram at emission stage; (**c**) the timing diagram at compensation stage.

**Figure 4 micromachines-16-01325-f004:**
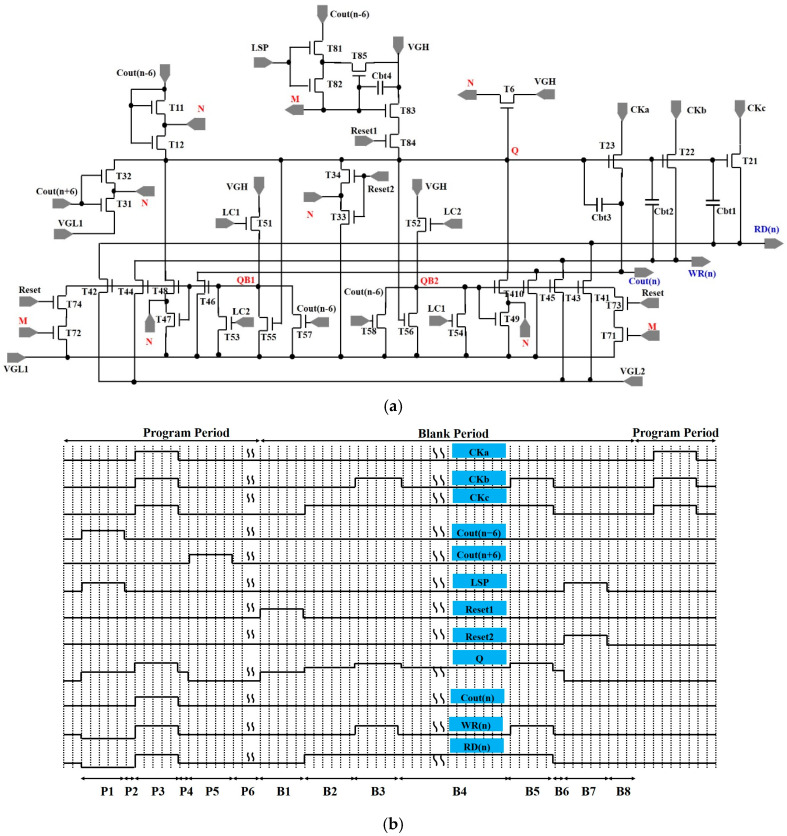
(**a**) A schematic and (**b**) the time diagram of the proposed GOA.

**Figure 5 micromachines-16-01325-f005:**
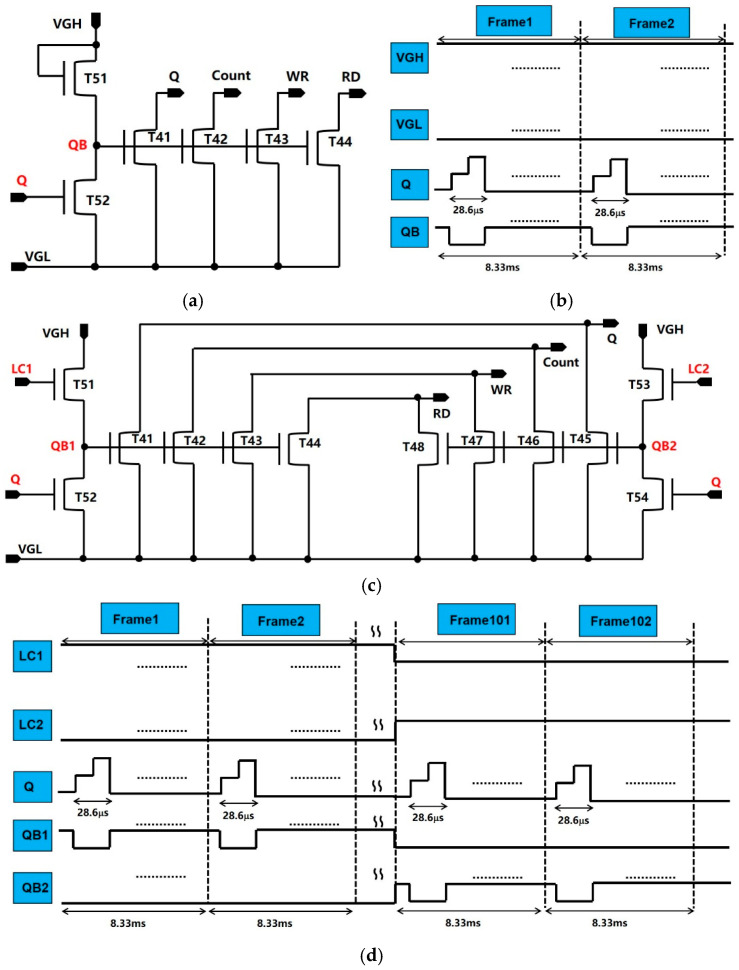
(**a**) A schematic of the conventional pull-down unit; (**b**) the time diagram of the conventional pull-down unit; (**c**) a schematic of the proposed pull-down unit; (**d**) the time diagram of the proposed pull-down unit.

**Figure 6 micromachines-16-01325-f006:**
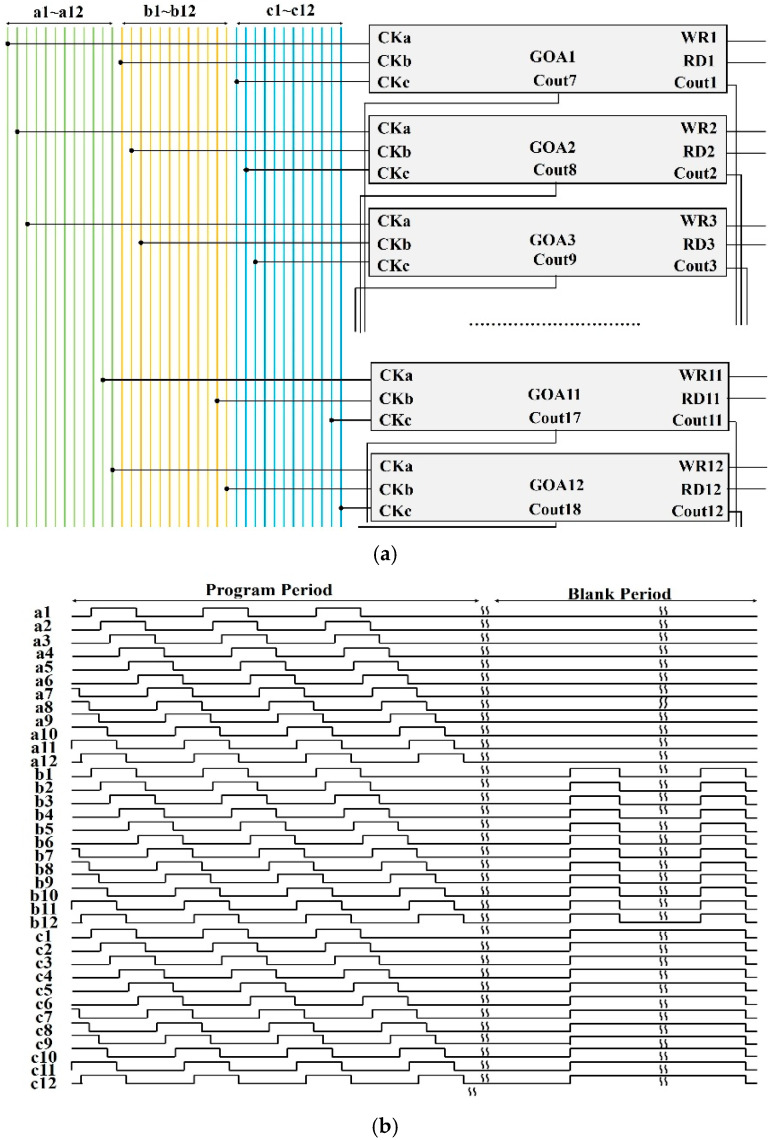
The diagrams of (**a**) GOA structure and (**b**) CK signals in the GOA.

**Figure 7 micromachines-16-01325-f007:**
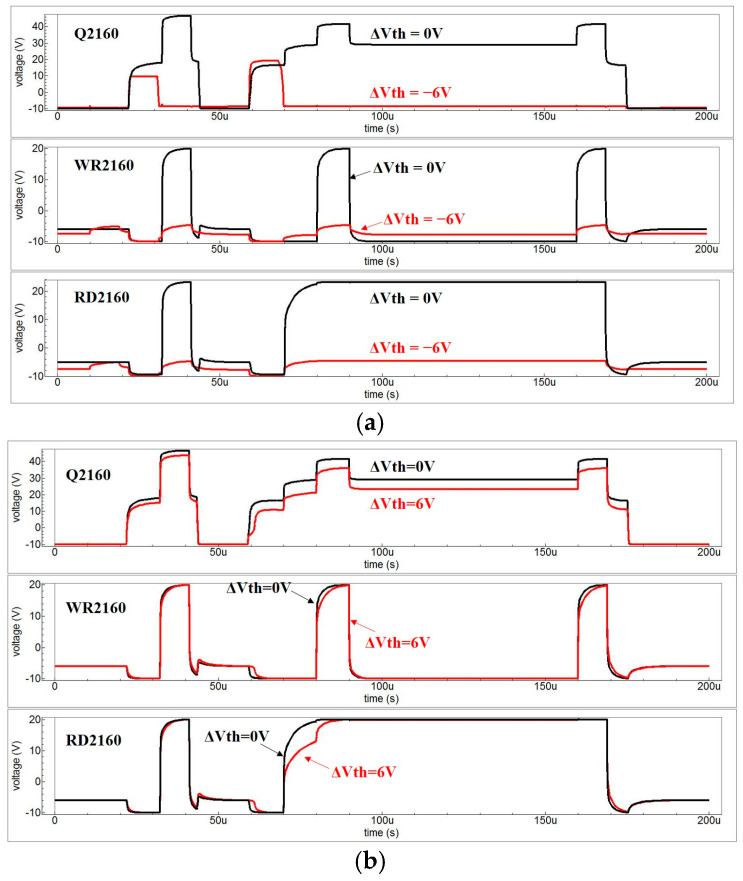
(**a**) the simulation result of output waveforms when Vth shifts by 0 V and −6 V; (**b**) the simulation result of output waveforms when Vth shifts by 0 V and 6 V.

**Figure 8 micromachines-16-01325-f008:**
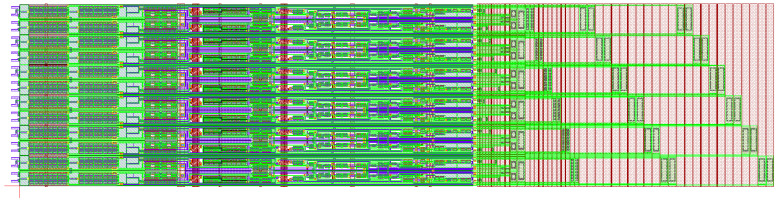
The detailed layout of GOA.

**Figure 9 micromachines-16-01325-f009:**
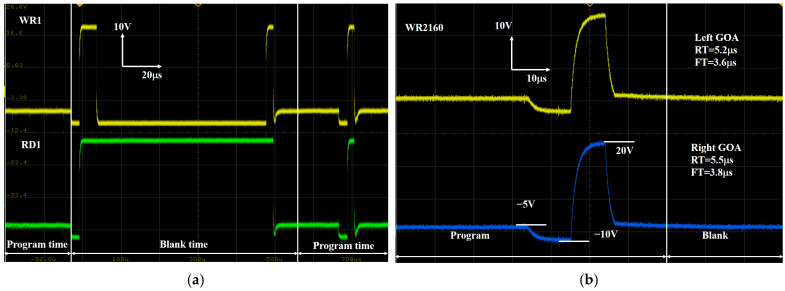
The GOA waveform (**a**) at the blank time for the first stage and (**b**) for the 2160th stage.

**Figure 10 micromachines-16-01325-f010:**
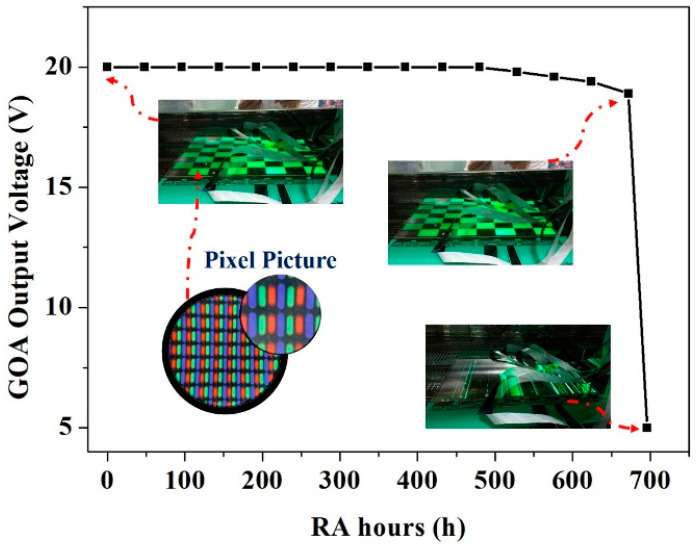
The output amplitude in GOA as a function of operation time.

**Figure 11 micromachines-16-01325-f011:**
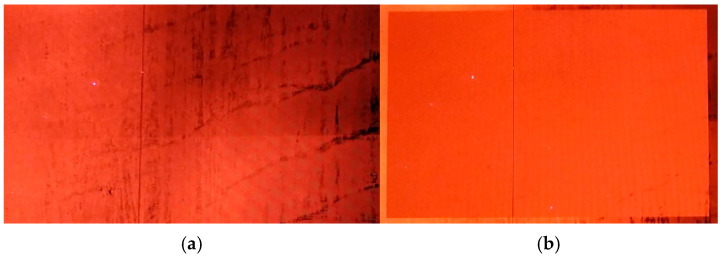
The display images (**a**) before compensation and (**b**) after compensation.

**Table 1 micromachines-16-01325-t001:** The parameters of devices in the GOA circuit.

TFT/Capacitance Size			
T11/T12/T21	500 μm/6.5 μm	T53~T56/T71~T74	50 μm/6.5 μm
T22/T23	2500 μm/6.5 μm	T57/T58	25 μm/6.5 μm
T31~T34/ T47~T410/ T81~T84	100 μm/6.5 μm	T6	15 μm/15 μm
T41~T44	300 μm/6.5 μm	Cbt4	2.5 pF
T45/T46	150 μm/6.5 μm	Cbt1/Cbt2/Cbt3	1.25 pF
T51/T52	10 μm/8 μm		

**Table 2 micromachines-16-01325-t002:** The amplitudes of input signals.

GOA Input Signals	
CKa1~CKa12/CKb1~CKb12/CKc1~CKc12	−10 V~20 V
LSP/Reset1/Reset2	−10 V~20 V
LC1/LC2/STV	−10 V~20 V
VGH	20 V
VGL1	−10 V
VGL2	−5 V

## Data Availability

The original contributions presented in this study are included in the article. Further inquiries can be directed to the corresponding authors.

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
