# Peer review of "A 31-Inch AMOLED Display Integrating a Gate Driver with Metal Oxide TFTs"

_micromachines, 2025, doi:10.3390/mi16121325_

Round 1
Reviewer 1 Report
Comments and Suggestions for Authors
The paper introduced a gate driver with pixel circuit for AMOLED. Overall quality of paper is appropriate, but following comments have to be addressed before publication.
1. In Fig. 3(a), the frame time is divided into the program phase and blank phase. However, in Fig. 3(b), after the programming of Rd[n], the circuit directly enters the blank phase without a distinct emission period. Therefore, it is unclear whether the proposed pixel driving method actually adopts a progressive emission (PE) scheme. Please clarify how the blank phase is defined within one frame time and what the driving scheme of the proposed circuit is.
2. Although the title of the paper emphasizes the gate driver circuit, the description and analysis of the gate driver operation are insufficient.
The manuscript mainly focuses on the external compensation method of the pixel circuit, while the explanation of the gate driver design and operation should be provided in more detail.
3. The paper states that the introduction of the two-group inverter unit and pull-down unit can reduce the PBTS effect and improve reliability. However, the explanation of how this structure mitigates PBTS is insufficient. Please provide a more detailed description or mechanism showing how the proposed circuit reduces the PBTS effect.
Author Response
Thank you so much for reviewing our work. We are delighted to be informed with a major review. The reviewers’ suggestions are very helpful to improve our manuscript. Here we submit our revised manuscript (the changes are highlighted in yellow) and point-to-point responses. We believe that we have improved the English writing.
Referee: 1
The paper introduced a gate driver with pixel circuit for AMOLED. Overall quality of paper is appropriate, but following comments have to be addressed before publication.
- In Fig. 3(a), the frame time is divided into the program phase and blank phase. However, in Fig. 3(b), after the programming of Rd[n], the circuit directly enters the blank phase without a distinct emission period. Therefore, it is unclear whether the proposed pixel driving method actually adopts a progressive emission (PE) scheme. Please clarify how the blank phase is defined within one frame time and what the driving scheme of the proposed circuit is.
Reply:The reviewer’s comment is very important for our work. Actually, the proposed pixel driving method adopts a progressive emission scheme. The description has been added in the revised manuscript.
Page 3, line 86-101
“The time diagram of convenient GOA can be seen in Fig.2a. The displays do not perform any special operations at the blank time. The pixels in each single row emit light immediately after writing the data voltage to the pixels in the system. Shift register pulses, separated by blank time at the end of each frame, transport to display area row by row [26-28]. As shown in Fig.2b, the progressive emission programming scheme has been used in our proposed system. The duration of blank time (Tblank) can be expressed as follows:
(1)
t (1.85ms) is the charging time of each pixel, h (2160) is the number of rows on the display and F (120HZ) is the frequency. The blank time (170ms) is calculated through the above formula and can be utilized to transmit closed captions, program rating information, time codes, and other data. The function of mobility compensation is to reduce the carrier mobility differences of TFTs in AMOLED pixels, thereby preventing the inconsistent light-emitting brightness among different pixels. The mobility compensation is performed during the blank time period to avoid affecting effective display and ensure real-time compensation, because these operations demand an independent timing window that does not interfere with display. ”
Page4, line 140-142
The blank time period is immediately adjacent to the display of the next frame for real-time compensation. Fig.3c indicates the procedure of mobility compensation at the blank time, which is divided into five stages:
Page 5, line 169-178
Vdata1 is written to the pixel in the next fame. The drain current of T1 in the next frame is expressed as
(10)
According the equations (9) and (10), the current is equal to the current in the previous frame after compensation, as shown in equation (11)
(11)
Therefore, the brightness non-uniformity caused by mobility differences can be promptly corrected by calculating the compensation value based on the data (Vdata0) from the previous frame and writing the simulated data (Vdata1) to the pixel in the next frame, preventing the accumulation of image deviations.
Referee: 2
Although the title of the paper emphasizes the gate driver circuit, the description and analysis of the gate driver operation are insufficient.
The manuscript mainly focuses on the external compensation method of the pixel circuit, while the explanation of the gate driver design and operation should be provided in more detail.
Reply:The reviewer’s comment is very important for our work. More details have been added in the revised manuscript Figure.5, Figure.6 and Figure.10.
Page 7-8, line 243-255
“To improve the reliability in GOA, an investigation to introduce two group inverter unit and pull-down unit is performed in this study. This is necessary to be able to decrease the positive bias temperature stress (PBTS) effect in the TFTs and improve electric reliability of TFTs [31]. The schematic of conventional pull down unit and the time diagram of conventional pull down unit can be seen in Fig.5a and Fig.5b, respectively. It can be seen that the voltage in QB almost maintains at high voltage. Therefore, T41~T44 suffer continuously PBTS during the working time. In this study, dual pull-down unit has been utilized in GOA circuits. The schematic of the proposed pull down unit and the time diagram of conventional pull down unit can be seen in Fig.5c and Fig.5d, respectively. The voltage of LC1 and LC2 switches between high and low for an interval of 100 frames. Alternately, T41~T44 and T45~T48 suffer PBTS during the working time. Therefore, the introduction of the two-group inverter unit and pull-down unit can reduce the PBTS effect and improve reliability.”
(a) (b)
(c)
(d)
Figure 5. (a) Schematic of conventional pull down unit; (b) time diagram of conventional pull down unit; (c) Schematic of proposed pull down unit; (d) time diagram of proposed pull down unit
Page 8, line 262-274
“Fig.6a and Fig.6b exhibit GOA diagrams and waveforms of CKs, respectively. Note that CK signals are relevant for GOA circuits in two ways. At first, the increasing number of CK lines has the benefit to reduce signal delay time in terms of the significant RC load-ing in displays with large sizes. More importantly, an overlapped structure in these CKs has been successfully developed to generate output waveforms in GOA circuits with high quality. Considering the balance between RC loading and rise time, the pulse widths of CK signals are settled to 8.9s during program time and the GOA circuits are driven by 36 CKs (a1~a12, b1~b12, c1~c12), which are supplied by the bypass pins from external source chips. In this display, the whole GOA circuits are composed of 181 basic loop units (12 GOA stages in a basic unit). It should be pointed out that an additional unit (the 181th unit), the output of which has to be disconnected to scan lines in the display area, is used to supply feedback signals for the 180th GOA unit. In addition, Cout(n-6) signals in the first 6 stages are driven by external start pulse.”
(a)
(b)
Figure 6. The diagrams of (a) GOA structure; (b) CK signals in GOA
Page 11, line 330-339
Figure 10. Output amplitude in GOA as a function of operation time
“Fig.10 shows the output amplitude as a function of stress time when GOA works under the condition of high temperature and humidity (60°C and 90% R.H.). The display continues to work with a checkerboard pattern. According to the display industry standard, a minimum operation time of 500h is required for mass production. GOA waveform indicates full swing output volt-age with the increasing operation time to 480h. Meanwhile, the display demonstrates high-quality chessboard image without any dark spots or shrinkages in pixels. Although the output amplitude deteriorates with increasing operation time to 672h, a high uniformity picture still can be obtained. Finally, when the operation time reaches 696h, the malfunction of GOA occurs, which is consistent with the picture of the display in this figure. Therefore, a maximum lifetime of 672h is obtained, demonstrating the high stability of GOA circuits.
- The paper states that the introduction of the two-group inverter unit and pull-down unit can reduce the PBTS effect and improve reliability. However, the explanation of how this structure mitigates PBTS is insufficient. Please provide a more detailed description or mechanism showing how the proposed circuit reduces the PBTS effect.
Reply:Thanks for the reviewer’s valuable question. More details have been added in the revised manuscript Figure.6 and Figure.10.
Page 7, line 243-255
“To improve the reliability in GOA, an investigation to introduce two group inverter unit and pull-down unit is performed in this study. This is necessary to be able to decrease the positive bias temperature stress (PBTS) effect in the TFTs and improve electric reliability of TFTs [31]. The schematic of conventional pull down unit and the time diagram of conventional pull down unit can be seen in Fig.5a and Fig.5b, respectively. It can be seen that the voltage in QB almost maintains at high voltage. Therefore, T41~T44 suffer continuously PBTS during the working time. In this study, dual pull-down unit has been utilized in GOA circuits. The schematic of the proposed pull down unit and the time diagram of conventional pull down unit can be seen in Fig.5c and Fig.5d, respectively. The voltage of LC1 and LC2 switches between high and low for an interval of 100 frames. Alternately, T41~T44 and T45~T48 suffer PBTS during the working time. Therefore, the introduction of the two-group inverter unit and pull-down unit can reduce the PBTS effect and improve reliability. ”
(a) (b)
(c)
(d)
Figure 5. (a) Schematic of conventional pull down unit; (b) time diagram of conventional pull down unit; (c) Schematic of proposed pull down unit; (d) time diagram of proposed pull down unit

Reviewer 2 Report
Comments and Suggestions for Authors
The manuscript proposed a technical scheme for compensating pixel mobility by integrating a logical sense unit into the GOA. This technical scheme is expected to enhance display performance, and it demonstrates certain practical application value. However, there have been related issues that need to be addressed as follows:
- In Section 2,why is the degree term of “Vs-Vref” a square in Equation (3)?
- In Section 2, what problem does the final formula illustrate? What information can be obtained from this formula?
- Line 151:the expression of the sentence "In contrast to convenient shift register timing, the important point of this study is the introduction of driving pulses at blank time, because procedure of program stage can’t be disturbed by the operation of mobility compensation" seems to be unclear.
- It is suggested that the author add more detailed and intuitive descriptions in Figure 4(a) to  enhance its readability.
- The titles of Section 3 and Section 4 are the same.
Author Response
Thank you so much for reviewing our work. We are delighted to be informed with a major review. The reviewers’ suggestions are very helpful to improve our manuscript. Here we submit our revised manuscript (the changes are highlighted in yellow) and point-to-point responses. We believe that we have improved the English writing.
Referee: 2
The manuscript proposed a technical scheme for compensating pixel mobility by integrating a logical sense unit into the GOA. This technical scheme is expected to enhance display performance, and it demonstrates certain practical application value. However, there have been related issues that need to be addressed as follows:
- In Section 2,why is the degree term of “Vs-Vref” a square in Equation (3)?
Reply:Thanks for the reviewer’s positive evaluation on our manuscript. We have made a mistake and corrected it.
Page 4, line 151, line 157
(4)
(6)
- In Section 2, what problem does the final formula illustrate? What information can be obtained from this formula?
Reply:Thanks for the reviewer’s valuable suggestions on our manuscript. The description has been added in the revised manuscript.
Page 5, line 169-178
“Vdata1 is written to the pixel in the next fame. The drain current of T1 in the next frame is expressed as
(10)
According the equations (9) and (10), the current is equal to the current in the previous frame after compensation, as shown in equation (11)
(11)
Therefore, the brightness non-uniformity caused by mobility differences can be promptly corrected by calculating the compensation value based on the data (Vdata0) from the previous frame and writing the simulated data (Vdata1) to the pixel in the next frame, preventing the accumulation of image deviations. ”
- Line 151:the expression of the sentence "In contrast to convenient shift register timing, the important point of this study is the introduction of driving pulses at blank time, because procedure of program stage can’t be disturbed by the operation of mobility compensation" seems to be unclear.
Reply:Thanks for the reviewer’s valuable suggestions on our manuscript. We have carefully rewritten the sentence.
For example
Page 3, line 86-101
“The time diagram of convenient GOA can be seen in Fig.2a. The displays do not perform any special operations at the blank time. The pixels in each single row emit light immediately after writing the data voltage to the pixels in the system. Shift register pulses, separated by blank time at the end of each frame, transport to display area row by row [26-28]. As shown in Fig.2b, the progressive emission programming scheme has been used in our proposed system. The duration of blank time (Tblank) can be expressed as follows:
(1)
t (1.85ms) is the charging time of each pixel, h (2160) is the number of rows on the display and F (120HZ) is the frequency. The blank time (170ms) is calculated through the above formula and can be utilized to transmit closed captions, program rating information, time codes, and other data. The function of mobility compensation is to reduce the carrier mobility differences of TFTs in AMOLED pixels, thereby preventing the inconsistent light-emitting brightness among different pixels. The mobility compensation is performed during the blank time period to avoid affecting effective display and ensure real-time compensation, because these operations demand an independent timing window that does not interfere with display. ”
Page 5, line 169-178
“Vdata1 is written to the pixel in the next fame. The drain current of T1 in the next frame is expressed as
(10)
According the equations (9) and (10), the current is equal to the current in the previous frame after compensation, as shown in equation (11)
(11)
Therefore, the brightness non-uniformity caused by mobility differences can be promptly corrected by calculating the compensation value based on the data (Vdata0) from the previous frame and writing the simulated data (Vdata1) to the pixel in the next frame, preventing the accumulation of image deviations. ”
- It is suggested that the author add more detailed and intuitive descriptions in Figure 4(a) to  enhance its readability.
Reply:Thanks for the reviewer’s valuable suggestions on our manuscript. The description has been added in the revised manuscript.
Page 7, line 243-255
“To improve the reliability in GOA, an investigation to introduce two group inverter unit and pull-down unit is performed in this study. This is necessary to be able to decrease the positive bias temperature stress (PBTS) effect in the TFTs and improve electric reliability of TFTs [31]. The schematic of conventional pull down unit and the time diagram of conventional pull down unit can be seen in Fig.5a and Fig.5b, respectively. It can be seen that the voltage in QB almost maintains at high voltage. Therefore, T41~T44 suffer continuously PBTS during the working time. In this study, dual pull-down unit has been utilized in GOA circuits. The schematic of the proposed pull down unit and the time diagram of conventional pull down unit can be seen in Fig.5c and Fig.5d, respectively. The voltage of LC1 and LC2 switches between high and low for an interval of 100 frames. Al-ternately, T41~T44 and T45~T48 suffer PBTS during the working time. Therefore, the in-troduction of the two-group inverter unit and pull-down unit can reduce the PBTS effect and improve reliability. ”
(a) (b)
(c)
(d)
Figure 5. (a) Schematic of conventional pull down unit; (b) time diagram of conventional pull down unit; (c) Schematic of proposed pull down unit; (d) time diagram of proposed pull down unit
Page 8, line 262-274
“Fig.6a and Fig.6b exhibit GOA diagrams and waveforms of CKs, respectively. Note that CK signals are relevant for GOA circuits in two ways. At first, the increasing number of CK lines has the benefit to reduce signal delay time in terms of the significant RC load-ing in displays with large sizes. More importantly, an overlapped structure in these CKs has been successfully developed to generate output waveforms in GOA circuits with high quality. Considering the balance between RC loading and rise time, the pulse widths of CK signals are settled to 8.9s during program time and the GOA circuits are driven by 36 CKs (a1~a12, b1~b12, c1~c12), which are supplied by the bypass pins from external source chips. In this display, the whole GOA circuits are composed of 181 basic loop units (12 GOA stages in a basic unit). It should be pointed out that an additional unit (the 181th unit), the output of which has to be disconnected to scan lines in the display area, is used to supply feedback signals for the 180th GOA unit. In addition, Cout(n-6) signals in the first 6 stages are driven by external start pulse.”
(a)
(b)
Figure 6. The diagrams of (a) GOA structure; (b) CK signals in GOA
- The titles of Section 3 and Section 4 are the same.
Reply:Thanks for the reviewer’s positive evaluation on our manuscript. We have made a mistake and corrected it.
- The operation of GOA circuit
- GOA circuit simulation and operation

Round 2
Reviewer 1 Report
Comments and Suggestions for Authors
Thank you for the author’s efforts, and all of the reviewer's questions have been fully addressed.